# Assessment of Nursing Workload as a Mortality Predictor in Intensive Care Units (ICU) Using the Nursing Activities Score (NAS) Scale

**DOI:** 10.3390/ijerph18010079

**Published:** 2020-12-24

**Authors:** Georgia Fasoi, Eirini C. Patsiou, Areti Stavropoulou, Evridiki Kaba, Dimitrios Papageorgiou, Georgia Toylia, Aspasia Goula, Martha Kelesi

**Affiliations:** 1Department of Nursing, School of Health and Care Sciences, University of West Attica, 12243 Athens, Greece; gfasoi@uniwa.gr (G.F.); astavropoulou@uniwa.gr (A.S.); dpapa@uniwa.gr (D.P.); gtoylia@uniwa.gr (G.T.); mkel@uniwa.gr (M.K.); 2Intensive Care Unit, General Hospital Korgialenio-Benakio E.E.S, 11526 Athens, Greece; diakoftoaigio@gmail.com; 3Department of Business Administration, School of Administrative, Economics and Social Sciences, University of West Attica, 12243 Athens, Greece; agoula@uniwa.gr

**Keywords:** intensive care units, workload, nurses, mortality, nursing activities score scale (NAS) scale

## Abstract

Introduction: Nursing activities score scales are valuable instruments for assessing the quality of nursing care provided in critically ill patients and easy to use in validating nurse staffing. The aim of this study was the assessment of nursing workload (NW) as a predictive factor of mortality by using the nursing activities score (NAS) scale. Materials and Methods: In this cross-sectional study of 91 days during 2019, convenience sampling was employed to recruit 82 registered nurses (RN) from three intensive care units (ICUs) of two public hospitals with 41 beds in total. Data were collected using the NAS scale, the researcher’s observation, the information given by the staff, and the nursing care reports. Descriptive and inductive statistics were used with significance level α = 0.05. The Statistical Package for Social Sciences (SPSS 25.0) was used for analyzing the data. Results: Females were the majority of the sample (84.1%), with a mean age of 38.9 (7.7) years, 87.8% had a nursing degree from a technological educational institute (T.E.I), the average working experience was 14 (8.1) years and the ICU experience was 12.9 (8.5) years. There were 3764 daily records of NAS with an average of 54.81 (2.34) and total NAS of 756.51 (150.27). The NW of the first day’s admission in the ICU was 65.15 (13.05), NW was 13.15 h/day and the NW of patients who died was 57.37 (4.06). The optimal nurse/patient ratio (NPR) was 1:1.82, while the existing NPR was 1:2.86. The mortality rate was 28.7%. Conclusion: Although the study results did not demonstrate a significant correlation between NW and mortality, the NW in ICU appeared to be relatively higher for patients who died than for those who survived. This result may serve as an indication for a positive correlation between these two variables. In addition, NW was found to be moderate, while one ICU nurse can take care of more than one patient.

## 1. Introduction

Worldwide, the development of modern models of nursing care follows the trend of the continuous upgrading of health services provided and of the documented evaluation of its results, through the design and implementation of scientifically accepted tools. The significant increase in the average age of the population, the available diagnostic and therapeutic tools, and the complexity that often characterizes a high-tech medical environment make health care a multidimensional phenomenon [1,2]. The evaluation of nursing workload (NW) is the subject of many studies, especially in recent years where health care providers try to reduce the cost of nursing and increase the quality of health care. Studies show that low nursing staffing rates are associated with a negative effect on patients’ health [3,4]. The development of NW monitoring and measurement systems is necessary to document the quality of health services provided, increase the professional satisfaction of nurses, and reduce the stress and burnout that are experienced frequently [5,6]. Nursing interventions in patients admitted in intensive care units (ICUs), if recorded on a daily basis, can be a predicting factor for patients’ outcomes and a tool for managing human and material resources effectively [7].

The nursing activities score (NAS) scale was developed in 2003 to measure the consumption of nursing time in ICUs, and is used as an evaluation tool for nursing workload, including measurements of clinical and administrative tasks performed by nurses [8]. The assessment of NW in adult ICUs with NAS scale has been the subject of many research studies at an international level. According to Lachance et al. [9], the NAS scale has been used in 34 studies, the majority of which were published in Brazil in the period between 2010 and 2014. This tool was used to evaluate the NW in ICUs, to compare the workload between units, and also to correlate it with the age of patients, length of stay (LOS), and mortality [9].

Padilha et al. [10], in a study of 200 patients treated in 4 different ICUs of a large Brazilian hospital, correlated NW through the NAS with various variables. They concluded that deceased patients were exposed to the NAS 2.65 times more than the rest. The average NAS in 19 adult ICUs in 7 countries was equal to 72.8%, with the highest score being 101.8% in Norway and the lowest, 45.5%, in Spain [10].

Gerasimou et al. [11] showed that there is a statistically significant positive correlation between NW, family satisfaction, and nursing care provided in ICUs. Finally, measurements revealed a lack of nursing staff in the morning shift, and NW was estimated to be 6 h and 51 min. Each employee had to work more than eight hours and fully justify the high levels of burnout reported by a large percentage of Greek nurses [11].

Gouzou et al. [2] for the first time in Greece measured NW in a cardiological ICU based on the NAS scale and was 41.23% (17.58) with a corresponding NPR of 1:2.5. NW showed a positive correlation with LOS (NAS r = 0.22, *p* < 0.05), and negative with the overall satisfaction of nurses (*p* < 0.05) [2,5]. Kiekkas et al. analyzed the effect of NW on the frequency of infections and patient mortality and found a positive correlation, underscoring the need for providing individualized patient care in a more coherent way [12]. Simões et al. [13] indicate that increased NW in ICUs is closely related to patients’ characteristics, such as gender, age, and severity of illness. They further suggest that knowledge of factors that influence NW in ICUs and effective management may contribute to the reduction of possible adverse events and improved care [13].

The aim of the present study was to measure NW in patients admitted to adult ICUs, as well as to correlate it with the mortality rate using the NAS scale. 

Additionally, the study intended to measure the nurses per patient ratio (NPR) required for the staffing of general adult ICUs, based on the workload that has been measured.

## 2. Materials and Methods

A cross-sectional study was conducted for a period of 91 days during 2019, regarding how NW relates to the mortality of ICU patients. A convenience sampling strategy was applied. The sample of the study consisted of 82 nurses working in 3 adult ICUs of 2 public hospitals in Athens including 41 beds in total, and this was the total number of nurses working on these units. There were 293 patients in the involved ICUs during the examined period. Data were collected by the researchers using the method of observation, collection of data from the nurses, and the nursing care reports. The NAS scale was used to collect data (Table 1). The NAS scale includes 23 items that measure the clinical and administrative tasks of nurses in ICUs. It calculates 81% of nursing time required to perform the interventions and is measured as a percentage of a nurse’s time from 0 to 177%. The resulting final score does not depend on the variety of cases and the type of ICU [8,14,15,16,17,18]. The recordings are approximately 24 h and data collection must be done at the same time each day, for each patient individually. It is possible to estimate the required staffing of an ICU, since 100 units of total NAS load represent 100% of nursing time/working hours. A NAS value of 50% indicates that a nurse can treat two patients simultaneously. A total NAS value of 350% means that during the period under consideration, 3.5 nurses must work in the ICU without interruption for a break [3,8,11,16,19]. Each grade of NAS corresponds to 14.4 min of work per 24 h and does not count time spent on non-nursing activities, such as personal activities (breaks or hand hygiene = 11.2% of working time), activities not directly related to the patient or not included in the health interventions, such as organizational issues, supervision of trainees, or supply of materials (= 6.3% of working time), and activities that are not included in the above categories (= 2.1% of working time). The sum of the time available for the above activities is estimated to correspond to 1 h and 34.08 min per 8-h shift [3,14]. The scale is considered valid and has been widely used internationally for the evaluation of NW in ICUs. The Greek version of the NAS scale has been validated by Gouzou et al. [2,5,20].

Permission for using the NAS scale was requested and the approval was granted prior to the commencement of the study. In total, 3764 measurements of NW were completed to calculate the NAS score. The demographic data of nurses were also recorded. Eighty-two registered nurses (RN) working in ICUs participated in the study. Each patient was assigned a code number. Gender, age, diagnosis of admission, length of stay (LOS) in the ICU, and disease outcome were recorded.

Ethical approval from the scientific committee of the two hospitals was obtained (protocol numbers: 498-30/09/2019, 29362-2/12/2019 and 348-09-01-2020) before the commencement of the study. The researchers preserved the basic principles of ethics in research as reported in the Helsinki Declaration. 

### Statistical Analysis

The data analysis was performed using the SPSS 25 (IBM, Athens, Greece). The level of statistical significance was set at 0.05. The statistical processing was based on the following steps: (a) In the descriptive part of the inductive statistics, data were recorded in all ICUs and per ICU. The mean values and standard deviations were calculated in the quantitative variables and the absolute and relative frequencies in the qualitative ones. (b) In the concluding part of the inductive statistics, the existence of differences between ICUs in terms of coverage of ICU beds with patients and the NW per patient was investigated, and for this purpose the analysis of variance (one-way ANOVA) was used. (c) Also, to investigate the relationship between NW/patient/NAS and survival variables (patients who survived and left the ICU), death, sex, age, and LOS, a correlation analysis was performed with the Pearson Correlation index. Cohen’s instructions were used to interpret the coefficient, which points out that the correlation with limits from 0.10 to 0.50+ is from small to large. (d) To check the effect of variables on patient outcome, a simple linear regression was performed.

Internal consistency reliability testing (Cronbach’s Alpha) was not applied to the original NAS scale because, according to Miranda et al. [8], this test is not required due to the construction of the tool. Recording some data automatically excludes the selection of others and opposes the notion of “internal cohesion”, since various elements of the scale do not all have the same chance of being selected. Therefore, the final internal cohesion factor in this tool appears to be incorrectly underestimated [8]. For this reason, no Cronbach’s Alpha test was included in the NAS validation results in the present study.

## 3. Results

Most of the nurses in the present study were women (84.1%), the mean age of the sample was 38.9 (7.7) years, and 87.8% were nurse graduates from T.E.I. with 30.5% having a master’s degree. The average years of working experience in the clinical area was 14 (8.1) and the average years of working experience in the ICU was 12.9 (8.5) (Table 2).

The number of patients in the present study was 293, with 65.2% being male. Diagnosis of admission was mostly associated with surgical causes. The mean age was 62.9 (17.71), average LOS in the ICU was 13.52 (13.15) days and the mortality rate was 28.7% (Table 3).

### 3.1. Descriptive Results of NAS Scale Measurements 

In total, 3764 NW measurements were performed per patient on the NAS scale. The tasks that were most frequently applied by nurses were recorded. In activities score item 1, 90.9% of the measurements required the presence of a nurse for more than two hours next to the patient’s bed. Regarding the activities score item 6 and the patient mobilization and change of position, in 93.4%, procedures were performed up to three times in 24 h, while in activities score item 8, in 92.8% of cases, nurses engaged in administrative and organizational tasks. In activities score items 9, 10 and 11, 99.9% of the patients needed respiratory support, 80.8% needed care of artificial airway care, and 88.1% underwent treatment to improve lung function (respiratory physiotherapy, spirometer breathing exercise, inhalation therapy, aspiration). Further results of the activities score items measured by the NAS scale are presented in Table 4. 

### 3.2. Descriptive Results of NW and Daily ICU Data

The total number of staff nurses working in all shifts per day according to the NAS should have been 7.57 (1.5) and the optimal NPR should have been 0.55 (0.07). Daily total NW/ICU was 756.51 (150.27), the mean value of NW/patient was 54.81 (2.34), and the NW of the first day’s admission in the ICU/patient was 65.15 (13.05). NW/patient was 13.15 h/day to take care of a patient. NW of patients who died was 57.37 (4.06), and for patients with a LOS of more than 20 days, NW was 55.41 (3.57). Further descriptive results of NW and daily ICU data are presented in Table 5.

### 3.3. Correlation of NW to ICU Patients’ Data

As shown in Table 6, existing NPR in the morning shift had a low negative correlation with NW according to the NAS scale (r = −0.225, *p* = 0.001), NPR in the afternoon shift was found to be moderate negative with NW according to the NAS scale (r = −0.450, *p* = 0.001), while existing NPR in the night shift was found to have a slightly negative correlation with NW according to the NAS scale (r = −0.261, *p* = 0.001).

As shown in Table 7, NW/patient was not found to be correlated with the gender and age of patients, while it was found to have low positive correlation with the NW of patients who survived and left the ICU (r = 0.159, *p* = 0.028) and with NW/patients who died (r = 0.364, *p* = 0.001). Still, it was found to have low positive correlation with the NW of patients with a LOS of up to 20 days (r = 0.385, *p* = 0.004) and with a LOS of more than 20 days (r = 0.267, *p* = 0.037). However, NW/patient was found to have a low negative correlation with the NW of patients with a LOS of up to 10 days (r = 0.267, *p* = 0.037). Total NW/ICU/24 h had low positive correlation with NW of patients with a LOS of up to 20 days (r = 0.399, *p* = 0.003), with a LOS of more than 20 days (r = 0.400, *p* = 0.001) and with death/day (r = 0.178, *p* = 0.003). The longer a patient is admitted to the ICU, the longer NW increases.

Aiming to investigate the effect of NW/patient who survived and left the ICU, a simple linear regression was performed. The NW of patients who survived and left the ICU was set as a dependent variable and NW/patient was set as an independent variable. The model revealed that if NW increases by 1 point, then the NW of patients who survived and were discharged will also increase by 0.523 (Table 8). 

NW Discharge of patients = 23.076 + 0.523 × NW

Aiming to investigate the effect of NW on patients who died in the ICU, a simple linear regression was performed. The NW of patients who died was set as a dependent variable and NW as an independent variable. The model revealed that if NW increases by 1 point, then the NW of patients who died will also increase by 0.791 (Table 9). 

NW Mortality of patients = 14.382 + 0.791 × NW

Aiming to investigate the effect of NW on patient deaths/day in ICUs, a simple linear regression was performed. Patient deaths/day were set as the dependent variable and NW/patient as the independent variable. The model revealed that if NW/patient increased by 1 point, then patient deaths/day would increase by 0.047 (Table 10). 

Patient deaths/day = −2.255 + 0.047 × NW

## 4. Discussion 

In the present study, 82 nurses participated and 3764 daily measurements of NW of NAS scale were performed in 293 patients. Independent variables of gender, age, level of education, years of working experience in clinical areas, and years of working experience in ICUs were not found to affect NW according to the NAS scale. All ICU nurses involved in the present study were graduates of either universities or T.E.I.s (tertiary education). 

Mean NW/patient was found to be 54.81 ± 2.34 for the NAS scale and this is similar to Reich et al.’s study [21] and lower compared to Gerasimou [11], where it was calculated at 73.7 ± 0.2, and Dede et al. [4], where it was 65.90 ± 7.19. High values appeared in Padilha et al.’s study [22], where mean NW was equal to 63.7 ± 2.4 [4,11,21,22]. It has been estimated that the average NAS value in 19 adult ICUs in 7 countries was 72.8 ± 31.1, with the highest value being 101.8 in Norway and the lowest, 45.5, in Spain [14,22]. Camuci et al. [23] measured NW in a burn unit and it was correspondingly high, similar to Nogueira et al.’s [15] study, in which NW was 71.3% and the diagnostic criteria for admission in ICUs (young adults with trauma) seemed to be correlated with NW. In Greece, NW has been calculated for cardiac patients, with the average value being equal to 41.23 ± 17.58, a value much lower than that of the present study. In addition, NW has been calculated for pediatric patients with an average value of 58.14, similar to the value of the present study [15,20,23,24,25].

Regarding NW of patients’ first-day admission in the ICU, it was 65.15 ± 13.05, where there is a significant increase in the average value of NAS. Similar values were found in Michali et al.’s study [3]. The average time of nursing care was calculated in minutes and this was found to correspond to 789.26 ± 33.7 min, a value much lower than in Dede et al.’s study [4], which was 948.89 ± 103.60 min. In Reich et al. [21], the time of immediate nursing care was estimated at 12.2 h for each patient within 24 h, a value less than the 13.15 h in the present study. In Oliveira et al. [26], it is reported that the lowest level of care required was approximately 14.9 h and the maximum was 19.5 h over a 24-h period. The above measurements were compared with those recommended by the Federal Council of Nursing (COFEN Resolution 293/2004) and correspond to the 17.9-h minimum average percentage of NAS that seems to belong to the established parameters. The present study therefore revealed a relatively low NW on a daily basis [3,4,21,26].

The measured number of nursing care resulted in the required number of nurses whose value was 7.57 (1.50) and the optimal NPR based on NAS was 1.82 (1.64–2.05), while the existing ratio was 2.86 (2.17–3.45). In Dede et al. [4], this number was much lower at 5.23 (0.70), and the optimal NAS based on NPR was 1.54 (1.40–1.65), while the current ratio was 2.30 (1.85–2.70). Existing NPR was associated with low to moderate negatives depending on the shift [4].

Regarding the analysis of the categories of the NAS scale, the activities that exceeded 90% were the use of drugs, the measurement of excreted urine, hygiene procedures, administrative duties, the presence of a nurse for 2 h by the bed, and the respiratory support; The findings agree with findings in previous studies [14,27,28]. Mortality was 28.7%, much higher than in other studies, where mortality ranged from 13.5% to 13.64%. Conversely, in Dede et al. [4] and Michali et al. [3], a higher mortality was observed (46.7% and 41%, respectively), and even higher in Kollia et al. [6], at the level of 71% [3,4,6,11,29].

65.20% of patients in this study were men. The mean age of patients in the study was 62.9 years, a value similar to Lucchini et al. [30], which was 60.97 years, while it was higher in Padilha et al. [22] and Dede et al. [4], where the mean age was 66.00 years. The above value, however, is higher than in other epidemiological studies evaluating the load, as in the study of Gerasimou et al. [11], where the mean age of patients was 59.21 (17.95) years and in the study by Goncalves et al. [14] the mean age was 53.7 (16.2) years. LOS was 13.52 (13.15) days, which is similar to other studies, where the mean LOS ranged from 4.6 to 17.58. In contrast, Michali et al. [3] revealed a high LOS of 24 (23) [1,3,4,11,15,22,30].

Bruyneel et al. [16] in French-speaking hospitals in Belgium showed that 64% of patients had been admitted with medical causes and had a NW of 68.6, which contradicts the present study, where the admission rate was 49.8% of surgical patients with a significantly lower NW [16]. Another important finding of this study was the statistically moderate difference between the mean daily NW/patient on the NAS scale between patients who died and those who survived. The average value of the NAS scale was 52.08 (6.85) for those who survived and 57.37 (4.06) for those who died. Increased, but similar, were the results in Dede et al. [4], where the mean value of NAS scale was 62.23 (6.38) for those who survived and 68.13 (5.12) for those who died, while in Italian ICUs the results were even higher, where the mean value of the NAS scale for survivors and the dead was found to be 63.88 (15.51) and 79.49 (21.46), respectively, and there was a statistically significant correlation between patient outcome and NW [4,28].

In addition, through the application of linear regression, it appeared that the NAS scale could be used as a predictor of patient outcome, as the model showed that if NW increased by 1 unit, the patient death would increase by 0.791 times, lower than Dede et al. [4], who revealed that patients with a mean NAS > 65.00 were 4.688 times more likely to die (*p* = 0.018) [4].

### Limitations of Study

The small sample in relation to the number of variables under this study is a significant limitation in terms of the ability to generalize the results to the population of ICU nurses. Also, according to the instructions of the NAS scale, the measurements took place once per day and some nursing procedures may have been omitted and not been integrated due to increased NW. An additional limitation of this study was the absence of an organized manner or system for measuring missed care or errors in the administration of medication in the involved sites, as both issues might be associated to NW and patient mortality. The development and application of such systems in ICUs is recommended as a future direction to improved monitoring and patient care. Finally, the predictions by means of a regression model must be taken with great caution since this study is descriptive, cross-sectional, and non-analytical.

## 5. Conclusions

This study tried to investigate the potential effect of nursing workload as a prediction of patient mortality. Nurse gender, level of education, and experience were parameters that were found to be not significantly correlated to nursing workload. NW in the ICU was measured and found to be close to average values, while one nurse can take care of more than one patient. NW in ICUs appeared to be relatively higher for patients who died than for those who survived, and this may serve as an indication for a positive correlation between these two variables.

A different methodology of measuring NW and its correlation to mortality, lack of a system for measuring missed care, heterogeneous populations under the study, as well as the severity of patients’ clinical condition admitted to ICUs, indicate the need for further research in the field.

## Figures and Tables

**Table 1 ijerph-18-00079-t001:** Nursing activities score items and weights.

a/a		SCORE
1	Monitoring and titration	
1a	Hourly vital signs regular registration and calculation of fluid balance.	4.5
1b	Present at bedside and continuous observation or active for ≥2 h in any shift for reasons of safety, severity, or therapy, such as noninvasive mechanical ventilation, weaning procedures, restlessness, mental disorientation, prone position donation procedures, preparation of administration of fluids, and/or medication.	12.1
1c	Present at bedside and continuous observation or active for ≥4 h in any shift for reasons of safety, severity, or therapy such as those examples listed earlier.	19.6
2	Extra laboratory measurements, biochemical investigations, and microbiologic investigations.	4.3
3	Medication: Vasoactive drugs excluded.	5.6
4	Hygiene procedures	
4a	Performing hygiene procedures such as dressing of wounds and intravascular catheters, changing linen washing patient incontinence, vomiting, burns, leaking wounds, complex surgical dressing with irrigation, special procedures (e.g., barrier nursing, cross-inflection related, room cleaning after infections, staff hygiene), etc.	4.1
4b	The performance of hygiene procedures took >2 h in any shift.	16.5
4c	The performance of hygiene procedures took >4 h in any shift.	20
5	Care of all drains except gastric tube.	1.8
6	Mobilization and positioning, including procedures such as turning the patient, mobilization of the patient, moving from bed to chair, team lifting (e.g., immobile patient, traction, prone position).	
6a	Performing procedure 1 time per 8 h.	5.5
6b	Performing procedure more frequently than 1 time per 8 h or with 2 nurses.	12.4
6c	Performing procedure with ≥3 nurses (and frequently).	17
7	Support and care of relatives and patient including procedures such as telephone calls, interviews, and counseling. Often, the support and care of either relatives or patient allow staff to continue with other nursing activities (e.g., communication with patients during hygiene procedures, communication with relatives while present at bedside and observing patient).	
7a	Support and care of relatives and patient requiring full dedication for approximately 1 h in any shift such as to explain clinical condition, dealing with patient and distress, difficulty, and family circumstances.	4
7b	Support and care of relatives and patient requiring full dedication for 3 h or longer in any shift such as to explain clinical condition, dealing with patient and distress, difficulty, and family circumstances.	32
8	Administrative and managerial tasks	
8a	Performing routine tasks such as processing of clinical data, ordering examinations, professional exchange of information (e.g., ward rounds).	4.2
8b	Performing administrative and managerial tasks requiring full dedication for approximately 2 h in any shift such as research activities, protocols in use, admission, and discharge procedures.	23.2
8c	Performing administrative and managerial tasks requiring full dedication for about ≥4 h of the time in any shift such as death and organ donation procedures, coordination with other disciplines.	30
9	Respiratory support: Any form of mechanical ventilation/assisted ventilation with or without positive end-expiratory pressure, with or without muscle relaxants; spontaneous breathing with positive endexpiration pressure (e.g., CPAP or biphasic positive airway pressure [biPAP]), with or without endotracheal tube; supplementary oxygen by any method.	1.4
10	Care of artificial airways: endotracheal tube or tracheostomy cannula.	1.8
11	Treatment for improving lung function: thorax psychotherapy, incentive spirometry, inhalation therapy, intratracheal suctioning.	4.4
12	Vasoactive medication, disregard type, and dose.	1.2
13	Intravenous replacement of large fluid losses. Fluid administration >3 L/m^2^/d, irrespective of type of fluid administrated. Hemofiltration techniques. Dialysis techniques.	2.5
14	Left atrium monitoring. Pulmonary artery catheter with or without cardiac output measurement.	1.7
15	Cardiopulmonary resuscitation after arrest; in the past period of 8 h (single precordial thump not included).	7.1
16	Hemofiltration techniques. Dialysis techniques.	7.7
17	Quantitative urine output measurement (e.g., by indwelling urinary catheter).	7
18	Measurement of intracranial pressure.	1.6
19	Treatment of complicated metabolic acidosis/alkalosis.	1.3
20	Intravenous hyperalimentation.	2.8
21	Enteral feeding: through gastric tube or other gastrointestinal route (e.g., jejunostomy).	1.3
22	Specific intervention(s) in the intensive care unit. Endotracheal intubation, insertion of pacemaker, cardioversion, endoscopies, emergency surgery in the past period of 8 h, gastric lavage. Routine interventions without direct consequences to the clinical condition of the patient, such as X-rays, echography, electrocardiogram, dressing, or insertion of venous or arterial catheters, are not included.	2.8
23	Specific interventions outside the intensive care unit. Surgery or diagnostic procedures.	1.9

**Table 2 ijerph-18-00079-t002:** Demographic characteristics of nurses in ICUs.

		*N*	%
Gender	Man	13	15.9
	Woman	69	84.1
Age	Average Age: 38.9 (7.7)		
Educational Level	University Graduates, MSc	10	12.2
	T.E.I. Graduates, MSc	25	30.5
	T.E.I. Graduates	47	57.3
Working Status	Permanent staff	57	69.5
	Fixed-term staff	25	30.5
Average years of working experience: 14 (8.1)	Average years of working experience in ICU: 12.9 (8.5)		

**Table 3 ijerph-18-00079-t003:** Demographic characteristics of patients and diagnosis upon admission in intensive care units (ICU).

	*N*	%	Mean	Diagnosis	*N*	%	LOS	*N*	%
Men	191	65.2		Pathological	125	42.7	−10 days	177	60.4
Women	102	34.8		Surgical	146	49.8	−20 days	55	18.8
Mortality	84	28.7		Trauma	22	7.5	−20+ days	61	20.8
Age	293	100	62.9 (17.71)						
<60 Years	106	36.2							
>60 Years	187	63.8							
Los	293	100	13.52 (13.15)						

**Table 4 ijerph-18-00079-t004:** Descriptive results of activities score items measured by the NAS scale.

a/a	*N*	%	a/a	*N*	%	a/a	*N*	%	a/a	*N*	%
1a	363	9.1	5	289	7.7	8c	0		16	132	3.5
1b	3421	90.9	6a	3515	93.4	9	3760	99.9	17	3764	100
1c	0		6b	249	6.6	10	3040	80.8	18	41	1
2	2684	71.3	6c	0		11	3317	88.1	19	32	0.8
3	3764	100	7a	1072	28.5	12	2721	72.3	20	1975	52.5
4a	3764	100	7b	0		13	84	2.2	21	2612	69.4
4b	0		8a	3495	92.8	14	0		22	0	
4c	0		8b	269	7.2	15	9	0.02	23	9	0.2

**Table 5 ijerph-18-00079-t005:** Descriptive results of NW and daily ICU data.

	*N*	Xmin	Xmax	Average	SD
NPR Morning	273	0.36	0.62	0.46	0.05
NPR Afternoon	273	0.21	0.50	0.31	0.07
NPR Night	273	0.21	0.40	0.29	0.04
Nurses Prediction	273	5.27	11.58	7.57	1.50
Optimal NPR	273	0.53	0.58	0.55	0.07
Beds	273	10	20.00	13.75	2.52
Total NW	273	527	1158.2	756.51	150.27
NW/Patient	273	48.8	60.98	54.81	2.34
NW In Minutes	273	702.72	878.11	789.26	33.7
1 Day NW	260	36.3	89.8	65.15	13.05
Survival	209	36.3	75.9	52.08	6.85
Death	84	48.6	67.7	57.37	4.06
Men	191	39.2	75.9	53.74	6.72
Women	102	36.3	73.4	53.43	6.55
Age Up To 60	106	40.40	73.4	54.36	6.01
Age > 60	187	36.30	75.90	53.14	6.92
Los −10	177	36.30	75.90	52.46	7.81
Los −20	55	45.60	66.00	54.90	4.27
Los −20+	61	45.90	63.50	55.41	3.57

**Table 6 ijerph-18-00079-t006:** Correlation of nursing workload (NW) per patient to the nurse/patient ratio (NPR).

		NPR Morning	NPR Afternoon	NPR Night
NW/Patient	Pearson Correlation	−0.225 **	−0.450 **	−0.261 **
	Sig. (2-tailed)	0.000	0.000	0.000
	N	273	273	273

** Correlation is significant at the 0.01 level (2-tailed).

**Table 7 ijerph-18-00079-t007:** Correlation between NW and ICU patients’ data.

Correlation		NW/Patient	Total NW
Survival	Pearson Correlation	0.159 *	−0.091
	Sig. (2-tailed)	0.028	0.209
	N	190	190
Death	Pearson Correlation	0.364 **	0.189
	Sig. (2-tailed)	0.001	0.086
	N	84	84
Men	Pearson Correlation	−0.083	−0.078
	Sig. (2-tailed)	0.252	0.283
	N	190	190
Women	Pearson Correlation	0.136	−0.058
	Sig. (2-tailed)	0.172	0.564
	N	102	102
Age < 60 Years	Pearson Correlation	0.029	0.107
	Sig. (2-tailed)	0.770	0.275
	N	106	106
Age > 60 Years	Pearson Correlation	−0.123	−0.054
	Sig. (2-tailed)	0.095	0.467
	N	185	185
Los −10	Pearson Correlation	−0.207 **	−0.143
	Sig. (2-tailed)	0.007	0.065
	N	167	167
Los −20	Pearson Correlation	0.385 **	0.399 **
	Sig. (2-tailed)	0.004	0.003
	N	55	55
Los −20+	Pearson Correlation	0.267 *	0.400 **
	Sig. (2-tailed)	0.037	0.001
	N	61	61
Deaths/Day	Pearson Correlation	0.195 **	0.178 **
	Sig. (2-tailed)	0.001	0.003
	N	273	273

** Correlation is significant at the 0.01 level (2-tailed); * Correlation is significant at the 0.05 level (2-tailed).

**Table 8 ijerph-18-00079-t008:** Nursing activities score (NAS) simple linear regression model and survival.

	Unstandardized Coefficients		Sig.	95.0% Confidence Interval for B	
	B	Std. Error		Lower Bound	Upper Bound
(Constant)	23.076	13.016	0.078	−2.601	48.752
NW	0.523	0.237	0.028	0.056	0.990

**Table 9 ijerph-18-00079-t009:** NAS simple linear regression model and mortality.

	Unstandardized Coefficients		Sig.	95.0% Confidence Interval for B	
	B	Std. Error		Lower Bound	Upper Bound
(Constant)	14.382	12.151	0.240	−9.790	38.554
NW	0.791	0.223	0.001	0.346	1.235

**Table 10 ijerph-18-00079-t010:** NAS simple linear regression model and patient deaths per day.

	Unstandardized Coefficients		Sig.	95.0% Confidence Interval for B	
	B	Std. Error		Lower Bound	Upper Bound
(Constant)	−2.255	0.785	0.004	−3.800	−0.711
NW	0.047	0.014	0.001	0.019	0.075

## Data Availability

Data sharing not applicable.

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
