# Peer review of "Assessment of Nursing Workload as a Mortality Predictor in Intensive Care Units (ICU) Using the Nursing Activities Score (NAS) Scale"

_ijerph, 2020, doi:10.3390/ijerph18010079_

Round 1

Reviewer 1 Report

The article is interesting for the results it provides, however it is not clarifying.

In order to carry out a more thorough review and understand all the data, it is necessary to carry out a review by the authors of the whole document, especially in the drafting of the results.

Authors must review and improve the document.

Basics: The mean and SD expression should be expressed Mean (SD) example 38.9(7.7), decimals must be separated by points, with commas extracting the thousands. Both forms are used throughout the manuscript 

In the introduction:" Nursing Activities Score scale (NAS) is one of the most up-to-date evaluation tool of NW and 46 was created in 2003 to evaluate nursing interventions" in this case it is necessary to cite the FRICE who published in 2003 the NAS.

Table 1 should be in the materials and methods section

The tables are not referenced in the text and it is necessary to relate text and tables, some of them are not understood as Table 4, the results out above are understood but the table is not since the Greek alphabet is used that in the previous description of the NAS (table 1) the latin alphabet is used.

There is a lot of numerical information in the text that is repeated in the tables, if the tables are expressed it is only necessary to influence the most important in the text, not repeat the data, so it is important that the tables are quoted in the text.

The results should be presented in a more orderly manner with titles that do not describe the analysis that has been done. They should be presented according to the objectives that respond to.

Reviewer 2 Report

In this study the authors investigate the potential effect of nursing workload as a prediction of patient mortality. The study involved 82 nurses and 293 patients from 3 ICUs in Greece. Nurse gender, level of education, experience were parameters also evaluated but were found not to be significantly correlated to nursing workload. The NAS scale was used as an evaluation tool for nursing workload which includes measurements of clinical and administrative tasks performed by nurses. Nursing workload / patient was measured to be close to average values when compared to most other published data. Also compared to similar studies the nursing workload in hours was found to be relatively low and there was moderate difference between the mean daily nursing load/patient between patient who died  and those who survived. Finally the authors apply linear regression for using NAS scale to predict patient outcome with regards to mortality and find a positive correlation between the two.

The statistical analysis results seem interesting; however, the following points need to be clarified before considering the study fit for publication.

  • Associating table 1 and table 4 is a bit complex as the numbering of NAS items is in English whereas the numbering used in Table 4 is in Greek.
  • X axis in figures 1 and 2 are labelled in Greek. Also the labelling of y axis in not clear.
  • Tables 2-5, 7-10 are not mentioned in the text
  • Is there a system in these 3 ICUs for measuring a missed care or medication administrations errors which may be associated to NW and patient mortality? If yes, can the authors include any information? If not, can they refer to this parameter in their discussion of limitation sections, as a future direction?

Reviewer 3 Report

Some comments are suggested:

  • In the abstract replace "last year" with the exact year.
  • In the abstract, the main results of NAS and the prediction of mortality have not been described in the results.
  • All keyword used should be MeSH descriptors.
  • Clarify all abbreviations the first time they appear (e.g. ICU)
  • The introduction of the manuscript is not really correct. The introduction should contain relevant information on the different similar studies carried out with NAS and the results obtained. In addition, the relationship between NAS and mortality from other studies has to be clearly seen as it is the objective of this research. However, what has been done in the introduction has been to describe the NAS. This should be in the measurement instruments of the methodology and not in the introduction.
  • The NAS validation results must also be included (Crombach's Alpha and others)
  • The university from which the nurse graduated is indicated as an inclusion criterion. This is confusing explained like this. An inclusion criterion is determined characteristics of the participants, which make them candidates for the study and which discriminate the sample. In this case, that's more of a sociodemographic characteristic.
  • Although it is a descriptive study, the universal population has not been explained and the sample size or without the total number of nurses in those units has been collected or what type of sampling has been carried out, if it was intentional or in what way.
  • It is indicated in the methodology that a simple linear regression has been carried out, which is somewhat risky as it is a cross-sectional study. Explain the use of that statistical test.
  • Tables should be self-explanatory and should include clarifications of the symbols used. For example, Table 4 is not understood.
  • The figures included do not provide much information and are not understood
  • In the conclusions, it is stated that an important correlation has been found between NW and mortality, when the correlations found in the results are quite low or moderate at best. They must downgrade the importance of the conclusion based on the results.
  • In the limitations include that the predictions by means of a regression model must be taken with great caution since it is a descriptive and non-analytical study.

Round 2

Reviewer 3 Report

The authors have included all comments and the article has been improved. Despite this, regarding the use of a linear regression, although they justify it and include it in the limitations, it is true that it is a statistical test that should be used in analytical studies that involve a comparison of control and experimental groups. Still, such an analysis could be accepted in this case.